# New Directions in Exercise Prescription: Is There a Role for Brain-Derived Parameters Obtained by Functional Near-Infrared Spectroscopy?

**DOI:** 10.3390/brainsci10060342

**Published:** 2020-06-03

**Authors:** Fabian Herold, Thomas Gronwald, Felix Scholkmann, Hamoon Zohdi, Dominik Wyser, Notger G. Müller, Dennis Hamacher

**Affiliations:** 1Department of Neurology, Medical Faculty, Otto von Guericke University, Leipziger Str. 44, 39120 Magdeburg, Germany; notger.mueller@dzne.de; 2Research Group Neuroprotection, German Center for Neurodegenerative Diseases (DZNE), Leipziger Str. 44, 39120 Magdeburg, Germany; 3Department Performance, Neuroscience, Therapy and Health, MSH Medical School Hamburg, University of Applied Sciences and Medical University, Am Kaiserkai 1, 20457 Hamburg, Germany; Thomas.Gronwald@medicalschool-hamburg.de; 4Department of Neonatology, Biomedical Optics Research Laboratory, University Hospital Zurich, University of Zurich, 8091 Zurich, Switzerland; felix.scholkmann@usz.ch (F.S.); dominik.wyser@hest.ethz.ch (D.W.); 5Institute for Complementary and Integrative Medicine, University of Bern, 3012 Bern, Switzerland; hamoon.zohdi@ikim.unibe.ch; 6ETH Zurich, Rehabilitation Engineering Laboratory, Department of Health Sciences and Technology, 8092 Zurich, Switzerland; 7Center for Behavioral Brain Sciences (CBBS), Universitätsplatz 2, 39106 Magdeburg, Germany; 8German University for Health and Sports, (DHGS), Vulkanstraße 1, 10367 Berlin, Germany; Dennis.Hamacher@dhgs-hochschule.de

**Keywords:** cognition, personalized training, personalized medicine, exercise prescription

## Abstract

In the literature, it is well established that regular physical exercise is a powerful strategy to promote brain health and to improve cognitive performance. However, exact knowledge about which exercise prescription would be optimal in the setting of exercise–cognition science is lacking. While there is a strong theoretical rationale for using indicators of internal load (e.g., heart rate) in exercise prescription, the most suitable parameters have yet to be determined. In this perspective article, we discuss the role of brain-derived parameters (e.g., brain activity) as valuable indicators of internal load which can be beneficial for individualizing the exercise prescription in exercise–cognition research. Therefore, we focus on the application of functional near-infrared spectroscopy (fNIRS), since this neuroimaging modality provides specific advantages, making it well suited for monitoring cortical hemodynamics as a proxy of brain activity during physical exercise.

## 1. Introduction

There is a robust body of literature suggesting that regularly conducted physical activity (typically engendered through regular physical exercise) promotes brain health and cognitive performance regardless of age [1,2,3,4,5,6,7,8,9,10,11]. However, the multilevel mechanisms driving exercise-induced neurocognitive changes across different age groups are not well understood [7,12,13] and little is known about which exercise prescription (e.g., intensity, duration, type of exercise) might be optimal to promote neurocognitive changes [1,7,8,14,15,16,17,18,19,20,21,22,23]. An adequate exercise prescription is the key to appropriately individualize physical exercise [15,24,25], but there is an ongoing debate about the optimal selection of parameters to do so [26,27,28,29,30,31,32,33,34,35,36]. This debate has now reached the field of exercise–cognition science [37,38,39] and could be a promising starting point to optimize exercise prescription.

To prescribe physical exercise, both parameters of external load (e.g., workload in Watts) and indicators of internal load (e.g., changes in heart rate) can be used. In this regard, external load is defined as work that an individual performs regardless of internal characteristics, whereas internal load encompasses the individual and acute psychophysiological responses to the external load, as well as influencing factors (environmental factors and lifestyle factors that amplify or diminish the physical exercise stimuli) [15,25,40,41,42,43,44,45,46,47]. In addition, it is important to emphasize that the external load has to be carefully adjusted to achieve a specific internal load. More importantly, specific indicators of internal load can be used as a proxy of the dose [15,25], which influences the effectiveness of an intervention [15,48,49]. However, it is currently not clear which indicator of internal load is the most suitable one to prescribe, for instance, exercise intensity in exercise–cognition science [15]. To extend this debate, we will discuss in this perspective article the role of brain-derived parameters as indicators of internal load and how these parameters serve to prescribe, rather than to solely monitor physical exercises (i.e., exercise intensity). Therefore, (i) we review which portable neuroimaging method would currently be the most suitable to monitor brain activity during physical exercise, (ii) we describe the neurophysiological background of this neuroimaging method and selected application-relevant methodological details, (iii) we explain the advantages of brain-derived parameters compared to conventional indicators of internal load (e.g., heart rate), and (iv) we outline potential opportunities for further investigation.

## 2. Which Portable Neuroimaging Tools Can Be Used to Assess Brain Activation During Physical Exercises?

Currently, there exist two portable neuroimaging modalities, namely electroencephalography (EEG) and functional near-infrared spectroscopy (fNIRS), to investigate brain activity during physical exercises in relatively unconstrained environments [50,51,52]. An overview on commercially available systems can be found in the review of Peake et al. [53]. Using EEG, brain activity is directly assessed by measuring electric changes in cortical layers [54]. Using fNIRS, brain activity is indirectly assessed by measuring cortical hemodynamic changes as a proxy of brain activity [50,55]. EEG has been applied in a variety of physical tasks and/or physical exercises (for review see [56]), such as balancing [57,58,59,60,61,62,63], walking [64,65,66,67,68,69], resistance exercises [70,71,72], or cycling [73,74,75,76,77,78,79]. Compared to fNIRS, EEG provides the advantage of a high temporal resolution (e.g., >1000 Hz) [80,81,82,83,84,85]. On the downside, EEG has the drawbacks of (i) a low spatial resolution (i.e., ~5 to 9 cm; with Laplacian transformation ~3 cm) [80,81,82,83,84,85,86,87], (ii) a time-consuming preparation when gel is used with wet electrodes [84,86], (iii) a high susceptibility to artefacts arising from motion, muscles, or sweat [80,81,87,88,89,90], and (iv) a hard interpretability of obtained signals for non-experts [91]. In contrast, fNIRS provides the advantages of a relatively high spatial resolution (i.e., ~1–3 cm) and a relatively high tolerance against motion artefacts [81,83,84,86,92,93,94,95,96,97,98], but suffers from a susceptibility to systemic physiological artefacts (e.g., superficial blood flow) [83,94,95,99,100,101]. Based on the drawbacks of EEG and the advantages of fNIRS, fNIRS is currently better suited for measurements of changes in cortical brain activity during physical exercises in unconstrained environments [50,102,103]. In fact, fNIRS has been applied during a variety of physical exercises such as juggling [104], balancing [105,106,107,108,109,110], walking (for review see [111,112]), resistance exercises [113,114,115,116], dancing [117,118,119], tai chi [120,121], climbing [122], synchronized swimming routines [123], table tennis [124], running [125,126,127], and predominantly during cycling [128,129,130,131,132,133,134,135,136,137,138,139,140,141,142,143,144,145,146,147,148,149,150,151,152,153,154,155,156,157,158,159,160,161,162,163,164,165]. Furthermore, fNIRS was used to monitor cerebral oxygenation during stationary cycling even in special cohorts, such as cardiac patients [166,167,168,169]. However, in the mentioned studies, fNIRS was only utilized to monitor brain activity during exercising, while, to our knowledge, no study so far has used fNIRS-based parameters to prescribe exercise variables (e.g., exercise intensity).

## 3. Neurophysiological Mechanisms and Physical Principles of fNIRS

The optical neuroimaging technique fNIRS allows the non-invasive imaging of cerebral hemodynamics from cortical layers in the human brain [83,111]. Neuroimaging based on fNIRS utilizes the physical principles of optical spectroscopy and the physiological processes of neurometabolic and neurovascular coupling [83], which are illustrated in Figure 1a. The execution of a distinct task (e.g., a motor–cognitive exercise such as dancing) causes a higher neural activation in specific brain regions. In order to supply the energy needed to satisfy the energetic demands of the activated neuronal tissue, the oxygen metabolism (neurometabolic coupling) is increased [83,170,171], leading to a higher metabolization rate of oxygen [170,171,172]. In consequence of the higher rate of oxygen metabolization, the local concentration of oxygenated hemoglobin (oxyHb) decreases and the local concentration of deoxygenated hemoglobin (deoxyHb) increases [170,171,172,173]. Furthermore, as shown in Figure 1a, an increase in neural activity also triggers local changes in cerebral hemodynamics and induces an intensified blood flow to the activated brain regions [83,170,174,175,176]. As a sequel of the locally increased blood flow, the local supply of oxygen is greater than its metabolization and, thus, a higher concentration of oxyHb and a decreased concentration in deoxyHb is to be observed in activated brain regions (see Figure 1a) [83,171,176]. These neural activity-dependent changes in oxyHb and deoxyHb concentrations can be used as indirect indicators of brain activation. To assess the neural activity-dependent changes in oxyHb and deoxyHb with fNIRS, light with distinct wavelengths in the near-infrared spectrum is emitted by a source (e.g., laser or light emitting diodes (LED)) on the scalp into the skull (see Figure 1b) [83,177,178]. Inside the skull, the emitted light travels through different layers (e.g., cerebrospinal fluid) and ideally penetrates the neural tissue [83,177,178]. In all the tissues of the head (intracerebral and extracerebral), the emitted near-infrared light undergoes scattering events and absorption processes, leading to light attenuation at the detectors [92,178,179]. Scattering events increase the length of their traveled photon paths as it forces the photons to deviate from their initial straight trajectories [83,92]. Absorption processes lead to a transformation of the initial energy of the photons into the internal energy of the respective medium (e.g., neural tissue) [83]. Based on the different absorption spectra of the chromophores (e.g., λ > 800 nm mainly oxyHb, λ < 800 nm mainly deoxyHb), neural activity-dependent changes in the local concentration of oxyHb and deoxyHb influence the local light absorption rate and, in turn, the regional magnitude of light attenuation [83,93,178]. These neural activity-dependent changes in light attenuation can be assessed by measuring the non-absorbed components of the emitted light using a detector that is placed on the head’s surface (see Figure 1b) [177,178]. Changes in light attenuation can be linked to local changes in cortical oxyHb and deoxyHb concentrations by means of the modified Beer–Lambert law, enabling a non-invasive quantification of the “indirect” indicators of brain activity changes [83,177,178].

With regard to the technical implementation of (functional) near-infrared spectroscopy ((f)NIRS), it is important to emphasize that the following four different methods exist. These different methods have unique advantages and disadvantages with respect to the usage in the field of exercise(–cognition) science: (i) continuous-wave NIRS (CW-NIRS), (ii) spatially resolved NIRS (SRS-NIRS), (iii) frequency domain NIRS (FD-NIRS) and (iv) time domain NIRS (TD-NIRS).
(i)In CW-NIRS devices, the changes in light intensity (i.e., attenuation) are used to calculate the relative concentration changes in chromophores (e.g., oxyHb and deoxyHb) [98,171,174,180,181]. Light with a distinct intensity is emitted into the tissue (e.g., brain and scalp tissue) via an emitter (e.g., placed on the scalp) and the non-absorbed light components leaving the tissue at a distinct point are measured via a detector, from which the intensity of outgoing light is obtained. By using CW-NIRS, changes in the attenuation coefficient can be calculated and used to determine the relative concentration changes in the chromophores (e.g., relative to baseline) [171,182].(ii)SRS-NIRS is a special type of CW-NIRS. In SRS-NIRS, at least two detectors (e.g., placed on the head surface) are used to measure the light which leaves the examined tissue after traveling through it (e.g., brain and scalp tissue) [183]. The information of the two detectors is used to determine the local gradients of light attenuation which, in turn, can be used to calculate the absolute concentration changes in chromophores (e.g., oxyHb and deoxyHb) and the tissue oxygenation index (TOI) [184,185]. TOI is the ratio of oxyHb to total hemoglobin (sum of oxy- and deoxyHb) and is also known as the tissue saturation index (TSI) and regional tissue oxygen saturation (StO_2_, and rSO_2_).(iii)In FD-NIRS, the source(s) (e.g., placed on the scalp) continuously emit(s) light with a distinct intensity into the tissue, whose amplitude is modulated at a specific frequency in the MHz range. A detector (e.g., placed on the head’s surface) measures the phase shift (delay) and light attenuation of the measured and non-absorbed light components, which, in turn, are used to determine the absorption and scattering properties of the specific tissue (e.g., brain tissue). Using the individual-specific information about the scattering and absorption properties of the distinct tissue allows the quantification of absolute concentration changes in the chromophores (e.g., oxyHb and deoxyHb) [98,171,174,180,181,186,187].(iv)In TD-NIRS, multiple sources emit extremely short light impulses into the tissue (e.g., brain and scalp tissue) and detectors, placed at a certain distance from the light emitting source, quantify the time of flight, the temporal distribution, and the shape of the temporal distribution of the non-absorbed photons, which leave the examined tissue (e.g., brain and scalp tissue). The information about the time of flight, temporal distribution, and shape of the temporal distribution of the non-absorbed photons are used to determine the scattering and absorption properties of the distinct tissue (e.g., brain tissue). In general, photons that travelled through the cerebral tissue are more delayed than photons that are only traveling through the scalp. The obtained information about scattering and absorption enables the calculation of absolute concentrations changes in chromophores (e.g., oxyHb and deoxyHb) [98,171,174,180,181,186,187].

With regard to the application of fNIRS in the field of exercise science (in particular to prescribe exercise), SRS-NIRS is the most promising of the four NIRS methods. SRS-NIRS allows, compared to CW-NIRS, the quantification of absolute changes in the concentration of chromophores [184,185], and, compared to FD-NIRS and TD-NIRS, SRS-NIRS devices are less expensive, have a higher acquisition rate, and are less bulky [98,180,188]. Furthermore, TD-NIRS suffers from the drawback that it is not able to detect small functional activation changes due to its working principle, which is based on time of flight photon detection. The latter returns a noisier parameter compared to SRS-NIRS, which is based on light attenuation measurement [171,180,188]. Moreover, multi-distance configurations of NIRS channels, as used in SRS-NIRS, allow measurements that are more robust against motion artefacts, resulting in a more stable acquisition of the signals [184]. In addition, in the field of exercise science, NIRS systems with LED sources (e.g., SRS-NIRS) are more suitable because they allow smaller and portable instrumentation and are safer in their application, as compared to NIRS systems with laser sources (e.g., FD-NIRS and TD-NIRS) [171,189]. More detailed information about the differences between CW-NIRS, SRS-NIRS, FD-NIRS, and TD-NIRS can be found in the referenced literature [98,171,174,180,181,183,185,186,190,191,192,193,194,195,196].

## 4. Advantages of Brain-Derived Indicators of Internal Load

In this section, we discuss the advantages of brain-derived indicators of internal load to prescribe physical exercise in exercise–cognition research. Therefore, we focus on the exercise variable *exercise intensity* because a full discussion of all exercise variables is beyond the scope of this perspective article. It is undoubted that the established approach prescribing the exercise intensity using specific indicators of internal load, such as percentages of maximal heart rate (HR_max_) or maximal oxygen uptake (VO_2 max_), which were obtained during a preceding graded exercise test, has its merits. However, this established approach for the prescription of exercise intensity can cause a considerable amount of interindividual heterogeneity in neurocognitive outcomes [15]. Hence, we recommend alternative approaches to exercise prescription, which make use of specific indicators of internal load, and which are known to be causally related to the intended outcome [15]. In this respect, fNIRS-derived brain parameters (e.g., oxyHb, deoxyHb, totHb, or StO_2_) can be a promising option for prescribing exercise intensity because they are more closely related to the organic system which is intended to be modulated (i.e., brain). This assumption is supported by the evidence outlined in the following:(a)cortical hemodynamics are sensitive to the level of physical load (e.g., exercise intensity) [125,126,128,139] and a decline in prefrontal oxygenation (i.e., oxyHb and StO_2_/TOI) at very high exercise intensities was observed [126,158,164,197,198]. The latter corroborates the notion of a “central governor” limiting maximal exercise performance [198,199,200,201,202,203,204,205],(b)cortical hemodynamics are sensitive to the influence of psychophysiological parameters (e.g., exercise tolerance) [129,206],(c)cortical hemodynamics during physical exercise are indicators of responsiveness, as levels of oxyHb in the right ventrolateral prefrontal cortex (PFC) during exercise are higher in individuals who show superior performance in a spatial memory task (e.g., being responders) after an acute bout of physical exercise [207],(d)the level of cortical hemodynamics during physical exercise might act as an indicator of the optimal brain state since lower levels of oxygenated hemoglobin in the PFC during physical exercise were associated with slower reaction times in an executive function task (i.e., Stroop task [208].

Regarding (c) and (d), comparable findings have, to the best of our knowledge, not been reported for conventional parameters of exercise prescription (e.g., heart rate (HR)) when neurocognitive outcomes were considered. Hence, (c) and (d) of the above-mentioned points especially support the idea that brain-derived parameters may be superior when prescribing physical exercise (e.g., exercise intensity) in the setting of exercise–cognition research. Moreover, commonly used conventional parameters of exercise prescription, such as HR, also suffer from the drawback that they are not able to sufficiently reflect psychophysiological responses to non-cardiorespiratory demands (e.g., cognitive load), which are posed by non-endurance exercises, such as simultaneous motor–cognitive exercise (e.g., dancing [14]). However, these demands are mirrored in brain-derived measures. Regarding fNIRS-derived brain parameters, it was reported that they are sensitive to:(e)the level of cognitive load [209,210,211,212],(f)the level of cognitive fatigue [213,214,215],(g)the influence of stress [216,217,218,219,220],(h)the influence of expertise level [104,122,212,221] or skill level [106,222],(i)training-related changes in motor–cognitive performance [105,117].

Accordingly, the provided evidence further buttresses the idea to use brain-derived parameters in exercise prescription. Brain-derived parameters, as compared to conventional parameters, have, at least theoretically, an added value in exercise prescription. Both the monitoring of brain parameters (e.g., during the course of a physical intervention) and the use of brain parameters to prescribe physical exercises (i.e., exercise intensity) are valuable options to gain more knowledge about the exercise–cognition interaction. The first approach (i.e., solely monitoring cerebral hemodynamics) can help us to answer specific research questions related to the effects of exercise on the brain which, in turn, can later be used to inform exercise prescription (e.g., *To what extent does an exercise prescription based on conventional parameters cause interindividual heterogeneity concerning cerebral hemodynamics?*). The second approach (i.e., brain-derived parameters to prescribe exercise) opens a new perspective, as it allows the study of the effects of an alternative exercise prescription on specific outcome measures (e.g., *To what extent does an exercise prescription based on brain-derived parameters influence interindividual heterogeneity in cognitive performance changes in response to an acute bout of physical exercise?*). Hence, both approaches should be seen as complementary options, which enable us to study the subject of exercise–cognition from two different perspectives.

## 5. Practical Implementation of Brain-Derived Parameters to Prescribe Physical Exercise

In the previous section, we have outlined that there is a strong theoretical rationale to use brain-derived parameters to prescribe exercise variables (e.g., exercise intensity), but this approach is currently underutilized. As this approach will open a new perspective on exercise–cognition interaction, we highlight potential areas of application by answering the following questions, which seem relevant for the practical implementation: *(i) How Can We Prescribe Exercise Intensity by Using fNIRS-Derived Brain Parameters?, (ii) Which Cortical Brain Area Should Be Targeted?, (iii) How Can We Minimize Confounders in Order to Successfully Apply fNIRS During Physical Exercise?* and *(iv) Which Additional Internal Load Indicators Should Be Recorded Alongside fNIRS?*

### 5.1. How Can We Prescribe Exercise Intensity by Using fNIRS-Derived Brain Parameters?

In our opinion, a prescription of exercise intensity using fNIRS-derived brain parameters can be implemented in a manner comparable to routinely used conventional indicators of internal load (HR). To test the practicability of the novel approach to prescribe exercise intensity by using fNIRS-derived brain parameters, continuous endurance exercises (e.g., cycling) could be a good starting point. Similar to the conventional approach of exercise prescription, the individual firstly performs a graded exercise test in which brain-derived indicators of internal load (e.g., StO_2_) can be measured alongside conventional ones (e.g., HR or VO_2_). Based on this graded exercise test, the external load (e.g., workload) can be identified, which corresponds, for instance, to the highest StO_2_ or to a specific percentage of a maximal StO_2_ (see Figure 2). This external load (i.e., workload) can then, in turn, be used to set the initial exercise intensity in an exercise session. Importantly, as fNIRS allows a non-invasive online monitoring of brain parameters, such as StO_2_ while exercising, the external load can be individually adjusted during the course of the exercise session, in order to ensure that the target StO_2_ is achieved (e.g., comparable to HR monitoring). Hence, using brain parameters, such as StO_2_, allows one to account for daily variations in performance capacity in the same manner as the established conventional approaches (e.g., heart rate monitoring). However, before the application of fNIRS can be recommended unreservedly, further research is necessary. Such research should aim, for instance, at investigating the reproducibility of fNIRS-derived brain parameters in the exercise setting, at studying the relationship between fNIRS-derived brain parameters and conventional parameters of exercise prescription (e.g., HR, level of peripheral blood lactate, and relative perceived exertion (RPE)), and at examining whether this novel approach of exercise prescription may be superior in inducing neurocognitive changes (e.g., acute influence on cognitive performance or blood-based markers), as compared to conventional approaches of exercise prescription.

### 5.2. Which Cortical Brain Area Should Be Targeted?

The exact placement of fNIRS optodes is crucial to obtaining cerebral hemodynamic signals from a specific brain area. To test the practicability of an exercise prescription which is based on brain-derived measures, we propose that the PFC could be a promising initial target area. The PFC is a key structure for the performance of cognitive control/executive functions [223,224], which is the most investigated cognitive domain in acute exercise–cognition studies [225]. More specifically, and based on evidence that is outlined in more detail below, the ventrolateral PFC, dorsolateral PFC, and the frontopolar area (FPA) are promising brain areas for investigations:(a)Regarding the ventrolateral PFC; it was observed that young adults with superior performance in a spatial memory tasks in response to an acute bout of endurance exercise (i.e., responders) exhibited higher levels of oxyHb in the right ventrolateral PFC during the exercise [207].(b)Regarding the dorsolateral PFC; it was noticed that in young adults, higher levels of oxyHb in the left dorsolateral PFC (measured during a cognitive test after exercising) [226,227,228] were associated with exercise-induced behavioral changes in the performance of the Stroop test.(c)Regarding the FPA; it was observed that in young adults, higher levels of oxyHb in the left FPA (measured during a cognitive test after exercising) [227] were associated with exercise-induced behavioral changes in the performance of the Stroop test. Similar findings have been observed for older adults regarding the right FPA [229].

In addition, more detailed information regarding an accurate and standardized placement of fNIRS optodes can be found in the referenced literature [176].

### 5.3. How Can We Minimize Confounders in Order to Successfully Apply fNIRS during Physical Exercise?

With respect to the practical application of fNIRS during physical exercising, it is mandatory to emphasize that movement artefacts or systemic physiological artefacts, which occur during physical exercising, influence the accuracy of the brain signal measurement of fNIRS significantly [99,101,149,151,230,231,232,233,234,235]. To minimize the influence of such signal confounders, appropriate data processing techniques should be applied (e.g., short-separation channel regression (SSR) to account for superficial blood flow [99,111,176,235,236] or SSR in conjunction with accelerometer signals to account for superficial blood flow and motion artefacts in CW-NIRS [237]). Furthermore, the application of other NIRS methods, namely SRS-NIRS, should be considered in the context of physical exercise, since they are less influenced by systemic physiological interference [238] and motion-related artefacts [184].

In addition, it is strongly recommended to record multiple physiological signals along with fNIRS signals (e.g., HR and heart rate variability (HRV), electrodermal activity (skin conductance), mean arterial blood pressure, systolic and diastolic blood pressure, respiration rate, and partial pressure of exhaled CO_2_ (P_ET_CO_2_)) to make valid assumptions about the physiological origin of the observed changes in fNIRS signals and, in turn, to improve the interpretation of fNIRS signals [99]. This approach has been recently termed systematic physiological augmented functional near-infrared spectroscopy (SPA-fNIRS) [239,240]. However, the study of Tempest et al. [108] supports the application of fNIRS, for instance, during motor–cognitive exercises, as robust cognitive task-evoked changes in cortical cerebral hemodynamics during cycling were measured by fNIRS, which are comparable to cortical hemodynamic changes observed without exercise [131]. Notwithstanding, it is important to emphasize that further efforts are necessary to improve the signal quality of fNIRS-based brain monitoring devices, which would allow for the more reliable conclusion of the origin of observed signal changes [99].

### 5.4. Which Additional Internal Load Indicators Should Be Recorded Alongside fNIRS?

In order to quantify the state of organic systems more comprehensively, it seems useful to complement the measures of the central nervous system (e.g., StO_2_ in the PFC) with (easily quantifiable) measures of the autonomous nervous system. A promising indicator of the state of the autonomous nervous system is the HRV, which is operationalized by the beat-to-beat variations of the heart rate over a specific time period (e.g., during a resting state or during physical exercises) [45,241,242,243,244,245]. HRV is considered to be a proxy of the actual health state [246,247,248,249] and stress level [247,250] of an individual. HRV also reflects, at least partly, the fitness level and daily readiness [243,244,251,252,253,254,255] of an individual, as well as the organismic demands [256,257,258,259,260]. There are numerous non-linear analysis approaches to appraise HRV, which reflect the systemic character of the organism [245,248,249,261,262,263]. Moreover, HRV is linked to cognitive performance in specific cognitive domains (e.g., during a resting state) [245,264,265,266,267,268,269,270,271,272,273] and it is, as outlined in the neurovisceral integration model, associated with the integrity of specific subareas of the PFC (e.g., medial PFC) [247,273,274,275,276,277,278,279,280]. In summary, this evidence suggests that HRV is a valuable parameter which should be recorded along with fNIRS in order to assess changes in the autonomic system more precisely. However, there is, to the best of our knowledge, no study available using HRV as an indicator of internal load to directly prescribe exercise intensity. Existing studies used resting-state HRV to guide the exercise prescription, but exercise intensity was prescribed with other parameters than HRV (e.g., HR or running velocity) [252,281,282,283].

Another parameter of interest is the pulse–respiration quotient, which provides a unique measure of the physiological state and traits [284,285] and which, therefore, should be more often assessed in physical exercise studies.

## 6. Summary and Conclusion

The optimal exercise prescription in the field of exercise–cognition is a topic of emerging interest, and a lively discussion on the matter has started. However, the question about which indicators of internal load may be the most appropriate ones remains open [15]. In this perspective article, we discuss and advocate for the use of brain-derived parameters in the exercise prescription of exercise–cognition studies (e.g., to prescribe exercise intensity). Brain-derived parameters provide the advantages that they are more closely related to the organic system that is aimed to be influenced and that they are sensitive to demands (e.g., cognitive load), which are not sufficiently reflected in conventional measures (e.g., HR). In particular, we discuss the promising potential of fNIRS-derived brain parameters (e.g., oxyHB, deoxyHb, totHb, and StO_2_) to prescribe physical exercise (e.g., exercise intensity) and we encourage the research community to test the practicability and effectiveness of this novel approach of exercise prescription.

## Figures and Tables

**Figure 1 brainsci-10-00342-f001:**
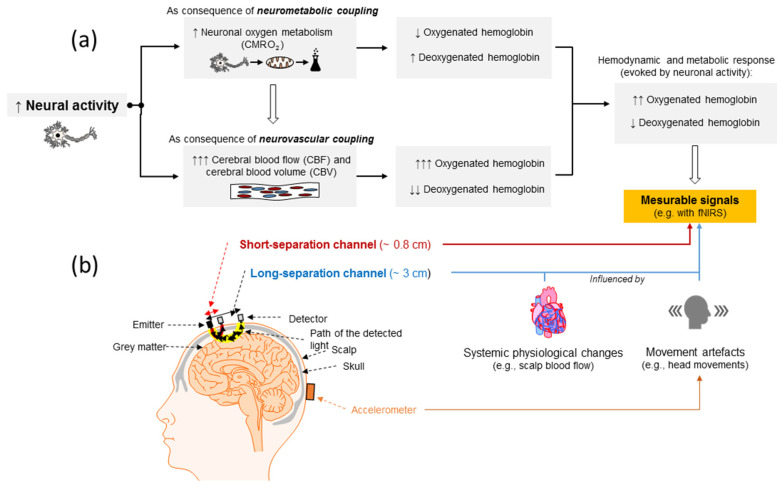
Schematic illustrations of (**a**) changes in cerebral hemodynamics and oxygen, induced by neural activity. (**b**) Depiction of a possible NIRS montage on the human head showing the assumed banana-shaped course of detected light of “short separation channels” and of “long separation channels”; fNIRS, functional near-infrared spectroscopy; CMRO_2_, cerebral metabolic rate of oxygen; ↑, increase; ↓, decrease.

**Figure 2 brainsci-10-00342-f002:**
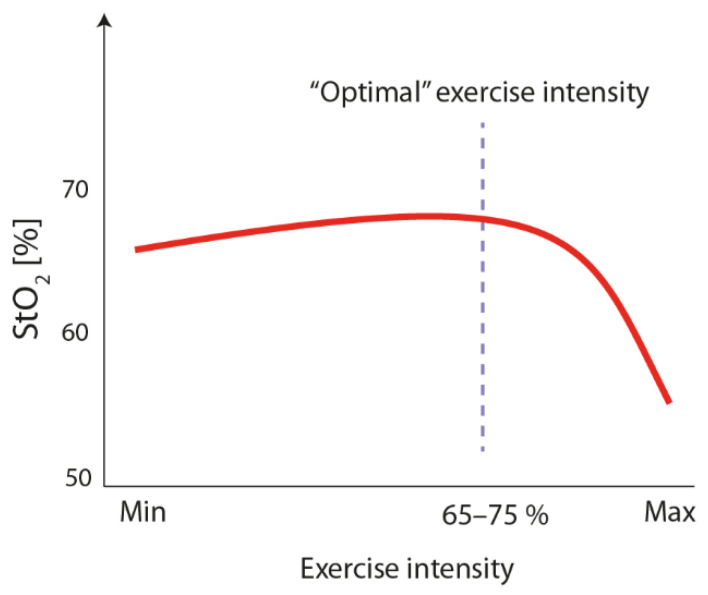
Schematic illustration of the time course of StO_2_ during a graded exercise test to exhaustion. The optimum is tentatively defined as the lowest exercise intensity leading to the highest cerebral oxygenation (marked by the dashed line).

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
