# Peer review of "New Directions in Exercise Prescription: Is There a Role for Brain-Derived Parameters Obtained by Functional Near-Infrared Spectroscopy?"

_brainsci, 2020, doi:10.3390/brainsci10060342_

Round 1

Reviewer 1 Report

Review brainsci-803515

Title: New directions in exercise prescription: Is there a role for brain parameters obtained by functional-near infrared spectroscopy?

The aim of this literature review is to discuss the role of brain measures as “novel” and valuable parameters of an individualized exercise and/or training prescription based on the example of motor-cognitive interventions.

Overall, the topic is novel and relevant for future training on cognitive-motor performance studies and the article is well written and very interesting! However, there are some aspects that need to be discussed and further reflected.

  1. The authors summarize their impressive literature overview with 361 references as a literature review. However, the article does not follow the main rules for a literature review (e.g. with respect to PRISMA guidelines).
  2. Common reviews give an overview of the quality of the included studies and describe their selection process. It remains unclear, if the authors give a full overview of this field or if they just pick up some prominent references. If there were no systematics in choosing the references (with inclusion and exclusion criteria) there might be a reporting/ selection bias. For example, the authors do not report a framework of cognitive-motor training from 2013 DOI 10.1007/s11556-013-0122-z in their section of cognitive motor training (page 7, line 207 ff. and especially the references in line 219 numbered with 186 and 187 seem to be wrong- they do not describe cognitive-motor interventions at all…).
  3. The different aspects of cognitive-motor improvement or the feasibility of the Mobile Brain Imaging (MoBI-approach) are not reported with respect to different age groups and settings. Therefore, it remains unclear, if the good ideas and thoughts of the authors can be used in general or if adaptations for different target groups are necessary. In the actual version the aspects are not reported with respect to e.g. children, adolescents, athletes or older adults. Can all study results be generalized for all target groups?  
  4. The authors discuss the feasibility of the portable fNIRS but do not report how their approach goes beyond the MoBI-approach.
  5. On page 2 line 46 the authors describe the idea of an appropriate training prescription and belonging aspects with own references. These ideas were already described in 2013 DOI 10.1007/s11556-013-0122-z and the authors should clarify that they made further suggestions to the work from 2013.
  6. It remains unclear why the authors use the term motor-cognitive instead of cognitive-motor. Most studies use the term cognitive-motor training with respect to e.g. cognitive-motor-interference.

My recommendations in summary:

Major

  • The authors should clarify that this is a narrative review and discussion paper and address the possible selection bias
  • The sections with cognitive-motor training and dose-response relationships should be sorted by different target groups or it should be mentioned for which target group these results are generalizable
  • The authors should better define in which regard their ideas go beyond previous framework of other research groups

Minor

  • The authors should proof the citations
  • The term motor-cognitive should be introduced with the synonymous use of cognitive-motor with this article or changed into cognitive-motor

Author Response

  • Reviewer 1

In this manuscript, (brainsci-803515) Herold and colleagues aim to establish a rationale for “the role of brain measures as ‘novel’ and valuable parameters of an individualized exercise and/or training prescription.” This is an interesting and nascent area, and I believe it is a topic worthy of exploration. However, this manuscript is poorly organized and there are a number of major flaws in the rationale that seem to contradict (or at least detract) from the title and stated aim of the review.

  • Thank you very much for valuable feedback. In the following we provide a point to point reply to your concerns. All changes in the revised manuscript were highlighted in “red”.
  1. First and foremost, it is not until Line 476 that the authors admit that much of the work necessary for any brain-derived measure of exercise dose has not yet been performed. This should be the message at the outset and a theoretically grounded discussion of that missing information and how it fits with our current understanding of exercise->cognition should follow.

  • We are thankful for your expert opinion. We have edited large parts of the manuscript and hope that it is now more suitable.

  1. This review and its organization would benefit from a brief synopsis of the cognitive domains exercise has the most potent effects on. For example, if there are consistently strong effects of exercise on cognitive control, then measuring exercise intensity in the appropriate circuits which contribute to function in that domain should be addressed. Right now, the manuscript generally treats the brain (and cognition) like a ‘black box’.

  • Thank you very much for your expert feedback. We have revised our manuscript and have incorporated this idea in section (ii) Which cortical brain area should be targeted?” (lines 290–310 in the revised version of the manuscript).

  1. The title suggests that the review will discuss ‘exercise’ prescription. However, Section 5 (lines 207-231) represents at best a series of translational errors and at worst a pedantic and ill-informed attempt at re-defining ‘exercise’. Please note my later point regarding Fissel’s framework, which could be used to motivate this paragraph, albeit with more accurate definitions

  • We are thankful for your constructive criticism. Accordingly, we have edited our manuscript and we focus now more on exercise prescription rather than on motor-cognitive exercise.

  1. The delineation between ‘exercise’ and ‘training’ is confusing at the outset, in the abstract (where the terms are initially conflated). Anyone trained in Kinesiology (or related fields) understands that the difference between physical activity and exercise lies in the intention of the movement—and it may be ultimately appropriate to reiterate that here. However, this makes the delineation between ‘exercise’ and ‘training’, and particularly the consistent use of ‘physical training’, to be quite confusing. In lines 210-212 it seems the authors are referring to differences between the effects of ‘acute exercise’ (a single bout) and ‘chronic exercise’ (or even ‘chronic exercise training’). I would recommend that the authors adopt this nomenclature, which is consistent with the recommendations of Budde et al., 2016 (cited as 195). Similar to the criticism leveled by Budde and colleagues in that letter, the current manuscript suffers from too many unnecessary distinctions and conflations of ‘exercise’: ‘physical training’, ‘endurance training’, ‘coordinative training’, ‘physical exercise’, ‘motor-cognitive intervention’, etc.

  • Thank you for your valuable feedback. We have simplified our wording and used the term “exercise” consistently.

The rest of the paragraph and rationale are only valid if the authors can give an example of an exercise that is void of any ‘cognitive task’. The problem begins on line 215: “we refer to motor as the physical part of a motor-cognitive exercise which is mostly characterized by its metabolic demands.” How is this possible and why is it necessary in the context of the stated aim of this article? Is this something that fNIRS alone can do? Dancing is described as ‘motor-cognitive with an incorporated cognitive task’, but cycling is only ‘cognitive’ if the individual is also performing arithmetic. What are the authors defining then as cognition?

  • We are thankful for pointing out this shortcoming. In the revised version of the manuscript we have removed this part.

So running does not have a cognitive component? Or is it that the authors are trying to describe a facet of exercise (i.e., ‘cognitive load’) that is important for understanding the exercise->cognition relationship?

  • Thank you for your constructive feedback. We agree with the reviewer and incorporate his thought in section “4. Advantages of brain-derived indicators of internal load” (lines 193-251)

Raichlen, D. A., Bharadwaj, P. K., Fitzhugh, M. C., Haws, K. A., Torre, G. A., Trouard, T. P., & Alexander, G. E. (2016). Differences in resting state functional connectivity between young adult endurance athletes and healthy controls. Frontiers in human neuroscience10, 610.

This paragraph is further muddled by the Netz (2019) review (cited as 202) which states “(1) Physical training affects cognition via improvement in cardiovascular fitness, whereas motor training affects cognition directly; (2) Physical training affects neuroplasticity and cognition in a global manner, while motor training is task-specific in increasing brain neuroplasticity and in affecting cognition.” Ignoring the outdated use of ‘physical training’, these points generally hold and may prove relevant for the topic at hand. However, this is not instantiated in the distinctions that the authors of the current manuscript are making.

  • We have removed this part of the manuscript.
  1. A basic tenet introduced in the abstract is that ‘exact knowledge about which exercise and/or training prescription is optimal (…) is lacking’. This sets a general tone for this paper that our inability to prescribe exercise accurately can be attributed to a lack of proper measurement, presumably (by the title) meaning that the authors will present fNIRS as a solution.
  • Indeed, we think that fNIRS-derived brain parameters could be a valuable option to prescribe, for instance, exercise intensity. We have outlined more clearly our rationale in the revised version of the manuscript.
  1. However, the authors seem to understand (e.g. Section 6) that the basic mechanisms for these benefits (e.g., exercise -> cognition) are also poorly understood, in part due to their multilevel complexity. Is fNIRS going to be used to prescribe exercise or can it help answer basic questions about the effects of exercise on the brain that will later inform prescription?
  • We appreciate your thoughtful comment and have incorporated it in lines 234-248 of the revised version of the manuscript.

Seemingly ignored is the field of systems neuroscience demonstrates that peripheral systems (sensitive to the effects of exercise training) are partially responsible for or mediate (in a mechanistic manner) brain function/organization. Some examples:

Mather, M., & Thayer, J. F. (2018). How heart rate variability affects emotion regulation brain networks. Current opinion in behavioral sciences, 19, 98-104.

Schiecke, K., Schumann, A., Benninger, F., Feucht, M., Baer, K. J., & Schlattmann, P. (2019). Brain–heart interactions considering complex physiological data: processing schemes for time-variant, frequency-dependent, topographical and statistical examination of directed interactions by convergent cross mapping. Physiological measurement, 40(11), 114001.

Schulz, S., Haueisen, J., Bär, K. J., & Voss, A. (2019). Altered causal coupling pathways within the central-autonomic-network in patients suffering from schizophrenia. Entropy, 21(8), 733.

Maric, V., Ramanathan, D., & Mishra, J. (2020). Respiratory Regulation & Interactions with Neuro-Cognitive Circuitry. Neuroscience & Biobehavioral Reviews.

Therefore, it is possible that adaptations in peripheral physiology (i.e., that we typically monitor to estimate exercise intensity) may play a role in the exercise->cognition pathway (see Thayer’s model of neurovisceral integration). To their credit, the authors mention the importance of monitoring systemic physiology (lines 515-527), but almost as an afterthought, without clear justification (“to quantify the state of the organ systems more precisely”), and without consideration for ‘bottom-up’ relationships. This mention also follows a paragraph (lines 501-515), which generally suggests that peripheral physiology should be treated as a nuisance variable or ‘noise’ in the context of the ‘brain’ NIRS signal.

  • Thank you again for your valuable hints. We have rewritten this part of the manuscript and incorporated the reviewer’s thought in section “(iv) Which other internal load indicators should be recorded alongside to fNIRS?”. We agree that peripheral physiology is important, and that a conventional exercise prescription has its merits (e.g., based on markers of peripheral physiology such as heart rate). The reviewer mentioned HRV (literature suggestions) and their relationship to neurocognition (e.g., Neurovisceral Integration Model). While we do not doubt that the autonomous nervous system has strong effects on the brain, HRV cannot be used to prescribe exercise intensity although it can be used to guide exercise intensity (see also lines 348-352 in the revised manuscript). However, in the case HRV is used to guide exercise intensity, the exercise intensity prescription still relies on mean HR or other parameters which are not that sensitive to changes in the brain. Hence, with this perspective article, we aim to put brain-derived measures in the focus of exercise prescription because this opens new perspectives to understand exercise-cognition interaction. By saying this, we see this novel approach as complementary to existing approaches (see lines 236-250 in the revised version of the manuscript).

  1. Section 6 makes very few jumps from ‘mechanism’ to ‘prescription’. The organization of this section is confusing and repetitive.
  • This section has been removed.

Lines 245-262: If ‘mismatch between supply/demand’ is a global mechanism, then why do we need to measure it in a small portion of the cortex during exercise? Is measuring the energy mismatch (presumably in HHb or TOI) in M1 or the SMA going to yield a different measure of intensity than measuring HR, RER, RPE, ventilatory equivalent of O2 etc? Please cite evidence.

  • We thank the reviewer for this thoughtful comment. With regard to the objection (“Please cite evidence”), we have to admit that there is currently no evidence available. To encourage the field to collect evidence in this direction, we have written this perspective article. We feel that it could be advantageous to test novel approaches of exercise prescription to move the field forward and gather new insights. By saying this, it would be interesting to investigate whether an exercise prescription based on StO2 in PFC or HR lead to a different prescription of exercise intensity. We incorporated this thought in section “(i) How we can prescribe exercise intensity by using fNIRS-derived brain?” (lines 262-288 in the revised version of the manuscript).

Line 279: It is the opinion of this reviewer that the guided plasticity facilitation framework needs to be introduced earlier in this manuscript be used as an organizational guide to discuss the utility of fNIRS to guide exercise prescription.

  • We thank the reviewer for this advice. As we have removed this part of the manuscript, we think that this comment is no longer applicable.

Lines 281-295: The focus on BDNF in this manuscript is odd. We know that there are extracerebral sources of BDNF, which means that blood-based measures of BDNF cannot reliably serve as a surrogate for BDNF that might ultimately modulate neurogenesis. Even then, BDNF is going to have to have the most striking effects on hippocampal structure and function and thus memory-specific cognition. The reviewers never focus on memory specifically, and this reviewer is not aware of any fNIRS technique capable of probing the hippocampus. Please consider striking or providing a rationale for singling out this mechanism linking exercise->cognition in the context of this review.

  • We have removed this part of the manuscript.

Line 296-310: ‘Guidance’ seems important for exercise interventions as part of a precision medicine approach in populations with dysfunctional or damaged brain networks. It is not clear how the activation of a diffuse motor network is going to change specific domains of cognitive function in healthy/intact individuals, much less how fNIRS will guide this prescription.

  • We do not completely understand the comment of the reviewer, but as we have edited large parts of the manuscript, this comment might be no longer applicable. Please, let us know if you have serious concerns about that. If you have serious concerns, we are willing to edit our manuscript accordingly.

By the end of this section this reviewer wonders if fNIRS should be in the title. The authors might consider removing it and also limiting Section 3 to a few sentences where relevant in Section 6.1.

  • We have edited large parts of the manuscript and pay a stronger focus on fNIRS and exercise prescription. Hence, we feel that the title fits to the revised version of the manuscript.

Additionally, there are a few minor oversights which should be addressed.

Line 79: The use of a surface Laplacian transformation can limit the surface fields to ~3cm about each electrode. Also, dense arrays allow for source-localization to be performed, which might be important for some of the mechanisms discussed in this article. Of course both techniques speak to the complexity of EEG processing, as the authors accurately point out.

  • We thank the reviewer for his expert opinion and have added this missing information (see line 81 in the revised manuscript).

Line 407-408: The authors have used a preponderance of space discussing different aspects of the exercise stimulus to this point, so it is not clear what aspect of the FITT principle has actually been left ‘beyond the scope’ of this article.

  • Thank you for this constructive feedback and pointing out this shortcoming. We focus in the revised manuscript on “exercise intensity” (see lines 192-194 in the revised version of the manuscript).

Line 428: Much focus on the PFC, but no link between BDNF (the favorite mechanism in this manuscript) and anything in the PFC. Also, localization matters. Dorsolateral PFC, venteromedial PFC, orbitofrontal PFC?

Thank you again for your valuable comments. We have addressed this point in section “(ii) Which cortical brain area should be targeted?” (lines 290-310 in the revised version of the manuscript).

Line 490: To earlier points: There are far more basic questions that need answered, or if they have been answered then they should be clarified here. Is TOI reliable enough for this? The differences between LEDs and lasers for this purpose was not discussed in Section 3, but is pertinent. Then, if the method is sufficient for this application, how does TOI relate to systemic VO2? If that hasn’t been answered then that is a starting point? Also, all measures in Line 486 are static—they are characteristics and thus not monitored during exercise.

  • Thank you for your “food for thoughts”. These points have been added in section “(i) How we can prescribe exercise intensity by using fNIRS-derived brain?” of the revised version of the manuscript (lines 262-288).
  • With respect to the objection of the difference between LED and lasers, we have added a sentence in section 3 (line 187-189).

Line 542: Unless the authors can present data supporting that fNIRS can measure endothelial shear stress in the cerebrovasculature, then this is not a sensible conclusion to draw here.

  • This part has been removed.

Reviewer 2 Report

In this manuscript, (brainsci-803515) Herold and colleagues aim to establish a rationale for “the role of brain measures as ‘novel’ and valuable parameters of an individualized exercise and/or training prescription.” This is an interesting and nascent area, and I believe it is a topic worthy of exploration. However, this manuscript is poorly organized and there are a number of major flaws in the rationale that seem to contradict (or at least detract) from the title and stated aim of the review.

  1. First and foremost, it is not until Line 476 that the authors admit that much of the work necessary for any brain-derived measure of exercise dose has not yet been performed. This should be the message at the outset and a theoretically grounded discussion of that missing information and how it fits with our current understanding of exercise->cognition should follow.
  2. This review and its organization would benefit from a brief synopsis of the cognitive domains exercise has the most potent effects on. For example, if there are consistently strong effects of exercise on cognitive control, then measuring exercise intensity in the appropriate circuits which contribute to function in that domain should be addressed. Right now, the manuscript generally treats the brain (and cognition) like a ‘black box’.
  3. The title suggests that the review will discuss ‘exercise’ prescription. However, Section 5 (lines 207-231) represents at best a series of translational errors and at worst a pedantic and ill-informed attempt at re-defining ‘exercise’. Please note my later point regarding Fissel’s framework, which could be used to motivate this paragraph, albeit with more accurate definitions
  4. The delineation between ‘exercise’ and ‘training’ is confusing at the outset, in the abstract (where the terms are initially conflated). Anyone trained in Kinesiology (or related fields) understands that the difference between physical activity and exercise lies in the intention of the movement—and it may be ultimately appropriate to reiterate that here. However, this makes the delineation between ‘exercise’ and ‘training’, and particularly the consistent use of ‘physical training’, to be quite confusing. In lines 210-212 it seems the authors are referring to differences between the effects of ‘acute exercise’ (a single bout) and ‘chronic exercise’ (or even ‘chronic exercise training’). I would recommend that the authors adopt this nomenclature, which is consistent with the recommendations of Budde et al., 2016 (cited as 195). Similar to the criticism leveled by Budde and colleagues in that letter, the current manuscript suffers from too many unnecessary distinctions and conflations of ‘exercise’: ‘physical training’, ‘endurance training’, ‘coordinative training’, ‘physical exercise’, ‘motor-cognitive intervention’, etc.

The rest of the paragraph and rationale are only valid if the authors can give an example of an exercise that is void of any ‘cognitive task’. The problem begins on line 215: “we refer to motor as the physical part of a motor-cognitive exercise which is mostly characterized by its metabolic demands.” How is this possible and why is it necessary in the context of the stated aim of this article? Is this something that fNIRS alone can do? Dancing is described as ‘motor-cognitive with an incorporated cognitive task’, but cycling is only ‘cognitive’ if the individual is also performing arithmetic. What are the authors defining then as cognition?

So running does not have a cognitive component? Or is it that the authors are trying to describe a facet of exercise (i.e., ‘cognitive load’) that is important for understanding the exercise->cognition relationship?

Raichlen, D. A., Bharadwaj, P. K., Fitzhugh, M. C., Haws, K. A., Torre, G. A., Trouard, T. P., & Alexander, G. E. (2016). Differences in resting state functional connectivity between young adult endurance athletes and healthy controls. Frontiers in human neuroscience, 10, 610.

This paragraph is further muddled by the Netz (2019) review (cited as 202) which states “(1) Physical training affects cognition via improvement in cardiovascular fitness, whereas motor training affects cognition directly; (2) Physical training affects neuroplasticity and cognition in a global manner, while motor training is task-specific in increasing brain neuroplasticity and in affecting cognition.” Ignoring the outdated use of ‘physical training’, these points generally hold and may prove relevant for the topic at hand. However, this is not instantiated in the distinctions that the authors of the current manuscript are making.

5. A basic tenet introduced in the abstract is that ‘exact knowledge about which exercise and/or training prescription is optimal (…) is lacking’. This sets a general tone for this paper that our inability to prescribe exercise accurately can be attributed to a lack of proper measurement, presumably (by the title) meaning that the authors will present fNIRS as a solution.

6. However, the authors seem to understand (e.g. Section 6) that the basic mechanisms for these benefits (e.g., exercise -> cognition) are also poorly understood, in part due to their multilevel complexity. Is fNIRS going to be used to prescribe exercise or can it help answer basic questions about the effects of exercise on the brain that will later inform prescription?

Seemingly ignored is the field of systems neuroscience demonstrates that peripheral systems (sensitive to the effects of exercise training) are partially responsible for or mediate (in a mechanistic manner) brain function/organization. Some examples:

Mather, M., & Thayer, J. F. (2018). How heart rate variability affects emotion regulation brain networks. Current opinion in behavioral sciences, 19, 98-104.

Schiecke, K., Schumann, A., Benninger, F., Feucht, M., Baer, K. J., & Schlattmann, P. (2019). Brain–heart interactions considering complex physiological data: processing schemes for time-variant, frequency-dependent, topographical and statistical examination of directed interactions by convergent cross mapping. Physiological measurement, 40(11), 114001.

Schulz, S., Haueisen, J., Bär, K. J., & Voss, A. (2019). Altered causal coupling pathways within the central-autonomic-network in patients suffering from schizophrenia. Entropy, 21(8), 733.

Maric, V., Ramanathan, D., & Mishra, J. (2020). Respiratory Regulation & Interactions with Neuro-Cognitive Circuitry. Neuroscience & Biobehavioral Reviews.

Therefore, it is possible that adaptations in peripheral physiology (i.e., that we typically monitor to estimate exercise intensity) may play a role in the exercise->cognition pathway (see Thayer’s model of neurovisceral integration). To their credit, the authors mention the importance of monitoring systemic physiology (lines 515-527), but almost as an afterthought, without clear justification (“to quantify the state of the organ systems more precisely”), and without consideration for ‘bottom-up’ relationships. This mention also follows a paragraph (lines 501-515), which generally suggests that peripheral physiology should be treated as a nuisance variable or ‘noise’ in the context of the ‘brain’ NIRS signal.

7. Section 6 makes very few jumps from ‘mechanism’ to ‘prescription’. The organization of this section is confusing and repetitive.

Lines 245-262: If ‘mismatch between supply/demand’ is a global mechanism, then why do we need to measure it in a small portion of the cortex during exercise? Is measuring the energy mismatch (presumably in HHb or TOI) in M1 or the SMA going to yield a different measure of intensity than measuring HR, RER, RPE, ventilatory equivalent of O2 etc? Please cite evidence.

Line 279: It is the opinion of this reviewer that the guided plasticity facilitation framework needs to be introduced earlier in this manuscript be used as an organizational guide to discuss the utility of fNIRS to guide exercise prescription.

Lines 281-295: The focus on BDNF in this manuscript is odd. We know that there are extracerebral sources of BDNF, which means that blood-based measures of BDNF cannot reliably serve as a surrogate for BDNF that might ultimately modulate neurogenesis. Even then, BDNF is going to have to have the most striking effects on hippocampal structure and function and thus memory-specific cognition. The reviewers never focus on memory specifically, and this reviewer is not aware of any fNIRS technique capable of probing the hippocampus. Please consider striking or providing a rationale for singling out this mechanism linking exercise->cognition in the context of this review.

Line 296-310: ‘Guidance’ seems important for exercise interventions as part of a precision medicine approach in populations with dysfunctional or damaged brain networks. It is not clear how the activation of a diffuse motor network is going to change specific domains of cognitive function in healthy/intact individuals, much less how fNIRS will guide this prescription.

By the end of this section this reviewer wonders if fNIRS should be in the title. The authors might consider removing it and also limiting Section 3 to a few sentences where relevant in Section 6.1.

Additionally, there are a few minor oversights which should be addressed.

Line 79: The use of a surface Laplacian transformation can limit the surface fields to ~3cm about each electrode. Also, dense arrays allow for source-localization to be performed, which might be important for some of the mechanisms discussed in this article. Of course both techniques speak to the complexity of EEG processing, as the authors accurately point out.

Line 407-408: The authors have used a preponderance of space discussing different aspects of the exercise stimulus to this point, so it is not clear what aspect of the FITT principle has actually been left ‘beyond the scope’ of this article.

Line 428: Much focus on the PFC, but no link between BDNF (the favorite mechanism in this manuscript) and anything in the PFC. Also, localization matters. Dorsolateral PFC, venteromedial PFC, orbitofrontal PFC?

Line 490: To earlier points: There are far more basic questions that need answered, or if they have been answered then they should be clarified here. Is TOI reliable enough for this? The differences between LEDs and lasers for this purpose was not discussed in Section 3, but is pertinent. Then, if the method is sufficient for this application, how does TOI relate to systemic VO2? If that hasn’t been answered then that is a starting point? Also, all measures in Line 486 are static—they are characteristics and thus not monitored during exercise.

Line 542: Unless the authors can present data supporting that fNIRS can measure endothelial shear stress in the cerebrovasculature, then this is not a sensible conclusion to draw here.

Author Response

Review brainsci-803515

Title: New directions in exercise prescription: Is there a role for brain parameters obtained by functional-near infrared spectroscopy?

The aim of this literature review is to discuss the role of brain measures as “novel” and valuable parameters of an individualized exercise and/or training prescription based on the example of motor-cognitive interventions.

Overall, the topic is novel and relevant for future training on cognitive-motor performance studies and the article is well written and very interesting! However, there are some aspects that need to be discussed and further reflected.

  • Thank you very much for valuable feedback. In the following we provide a point to point reply to your concerns. All changes in the revised manuscript were highlighted in “red”.

  1. The authors summarize their impressive literature overview with 361 references as a literature review. However, the article does not follow the main rules for a literature review (e.g. with respect to PRISMA guidelines).

  • Thank you for this advice, we have added in the abstract that this article is a “perspective article”.

  1. Common reviews give an overview of the quality of the included studies and describe their selection process. It remains unclear, if the authors give a full overview of this field or if they just pick up some prominent references. If there were no systematics in choosing the references (with inclusion and exclusion criteria) there might be a reporting/ selection bias. For example, the authors do not report a framework of cognitive-motor training from 2013 DOI 10.1007/s11556-013-0122-z in their section of cognitive motor training (page 7, line 207 ff. and especially the references in line 219 numbered with 186 and 187 seem to be wrong- they do not describe cognitive-motor interventions at all…).

  • Thank you for this advice. To make the readers aware that this is no systematic review, we have added in the abstract that this article is a “perspective article” (line 33).
  • As we have changed the focus of the manuscript to exercise prescription, the second objection of a missing framework of motor-cognitive training is no longer applicable.
  • We have proofed the citation. Indeed, in the old version of the manuscript a mistake occurred. In the new version of the manuscript, this part has been removed.

  1. The different aspects of cognitive-motor improvement or the feasibility of the Mobile Brain Imaging (MoBI-approach) are not reported with respect to different age groups and settings. Therefore, it remains unclear, if the good ideas and thoughts of the authors can be used in general or if adaptations for different target groups are necessary. In the actual version the aspects are not reported with respect to e.g. children, adolescents, athletes or older adults. Can all study results be generalized for all target groups?  

  • We are thankful for the advice of the reviewer. To focus more on exercise prescription (as suggested by the other reviewer), we have rewritten large parts of the manuscript and have removed most parts about motor-cognitive interventions. Hence, we think that this comment is no longer applicable.

  1. The authors discuss the feasibility of the portable fNIRS but do not report how their approach goes beyond the MoBI-approach.

  • Thank you again for your constructive criticism. We have edited our manuscript and in the current version we focus more on exercise prescription rather than motor-cognitive exercise. Our approach differs from the MoBI-approach as we propose that fNIRS-derived brain parameters can be used to prescribe exercise intensity rather than only monitor it (as in MoBI-approach).

  1. On page 2 line 46 the authors describe the idea of an appropriate training prescription and belonging aspects with own references. These ideas were already described in 2013 DOI 10.1007/s11556-013-0122-z and the authors should clarify that they made further suggestions to the work from 2013.

  • Thank you for this advice. We have incorporated this reference.

  1. It remains unclear why the authors use the term motor-cognitive instead of cognitive-motor. Most studies use the term cognitive-motor training with respect to e.g. cognitive-motor-interference.

  • Thank you for your expert opinion. However, as we have changed the focus of the manuscript, we think that this comment is no longer applicable.

My recommendations in summary:

Major

  • The authors should clarify that this is a narrative review and discussion paper and address the possible selection bias

  • Thank you for this hint. We have clarified in the introduction section that this is a “perspective article”.

  • The sections with cognitive-motor training and dose-response relationships should be sorted by different target groups or it should be mentioned for which target group these results are generalizable

  • We greatly appreciate your feedback, but as we have changed the focus of our perspective article to better illustrate our ideas regarding the use of fNIRS-derived parameters in exercise prescription, we feel that this comment is no longer applicable.

  • The authors should better define in which regard their ideas go beyond previous framework of other research groups

  • Thank you again for your constructive criticism. We have changed the focus of our manuscript to present our idea to use fNIRS-derived parameters to prescribe exercise intensity more prominently.

Minor

  • The authors should proof the citations

  • Many thanks for the careful proofreading. We have proofed the citations. Indeed, in the old version of the manuscript a mistake occurred. In the new version of the manuscript, this part has been removed.

  • The term motor-cognitive should be introduced with the synonymous use of cognitive-motor with this article or changed into cognitive-motor

  • We are thankful for your expert advice. However, as we have changed the focus of our manuscript, we feel that this comment is no longer applicable.
